# A Critical Assessment of the Association between HLA-G Expression by Carcinomas and Clinical Outcome

**DOI:** 10.3390/ijms22158265

**Published:** 2021-07-31

**Authors:** Ricky B. van de Water, Daniëlle Krijgsman, Ruben D. Houvast, Alexander L. Vahrmeijer, Peter J. K. Kuppen

**Affiliations:** 1Department of Surgery, Leiden University Medical Centre, P.O. Box 9600, 2300 RC Leiden, The Netherlands; r.b.van_de_water@lumc.nl (R.B.v.d.W.); R.D.Houvast@lumc.nl (R.D.H.); A.L.Vahrmeijer@lumc.nl (A.L.V.); 2Molecular Cancer Research, Centre for Molecular Medicine, University Medical Centre Utrecht, P.O. Box 85060, 3508 AB Utrecht, The Netherlands; D.Krijgsman-4@umcutrecht.nl

**Keywords:** HLA-G, immunotherapy, carcinoma, clinical outcomes

## Abstract

Human leukocyte antigen-G (HLA-G) conveys immunological tolerance at the maternal-foetal interface. HLA-G expression by tumour cells may also play such a role, resulting in tumour immune evasion, making HLA-G a potential target for immunotherapies. The aim of this review was to determine to what extent it is justified that HLA-G expression is considered as a target for immune checkpoint inhibiting therapy by critically assessing the association between HLA-G expression by carcinomas and clinical outcome of patients. The used HLA-G-detecting mAb, HLA-G quantification methods and statistically significant HLA-G-associated clinicopathological parameters are discussed. Tumour HLA-G expression correlated with poor clinical outcome in breast, esophageal, gastric and hepatocellular carcinoma patients. Tumour HLA-G expression was not associated with clinical outcome in ovarian and oral carcinoma patients. Cervical, colorectal, lung, and pancreatic carcinoma patients presented discrepant and therefore inconclusive results regarding the association between tumour HLA-G expression and clinical outcome. These disparities might partly be the result of differences in the methodological approach to quantify HLA-G expression between studies. Therefore, implementation of universal methodological procedures is strongly advised. Overall, HLA-G expression did not univocally result in poor clinical outcome of carcinoma patients. This implies that tumour HLA-G expression is not necessarily part of an inhibited tumour-immune response and tumour progression. Consequently, it remains elusive whether HLA-G expression by carcinomas functions as an immune checkpoint molecule affecting a tumour-immune response. It may also reflect derailed control of gene expression in tumours, with no real functional consequences.

## 1. Introduction

Human leukocyte antigen G (HLA-G) is a non-classical HLA-class I molecule which is exclusively expressed on the surface of preimplantation embryos and extravillous trophoblasts (EVTs) in the placenta at the maternal-foetal interface [1,2,3]. Here, HLA-G aides in creating an immunosuppressive environment, thereby establishing maternal immunological tolerance towards the foetus [1,3]. The immunosuppressive capabilities of HLA-G are achieved via interaction of HLA-G with its receptors Ig-like transcript 2 (ILT2), ILT4, and killer cell immunoglobulin-like receptor 2DL4 (KIR2DL4), which are presented differentially on specific immune cells [4,5,6].

The full-length HLA-G transcript is composed of a heavy-chain consisting of three globular domains (α1, α2 and α3), a signal peptide, and a transmembrane domain with a cytoplasmic tail [7]. The full-length mRNA transcript translates into the HLA-G1 isoform. At least six other isoforms (HLA-G2 to HLA-G7) can be generated through alternative splicing [8]. HLA-G1, -G2, -G3 and -G4 are membrane-bound isoforms due to retainment of their transmembrane domain. This transmembrane domain is not present in isoforms HLA-G5, -G6 and -G7 due to a stop codon in intron 4 for HLA-G5 and -G6, and in intron 2 for HLA-G7 [9]. Additionally, HLA-G1 and -G5 isoforms are able to non-covalently bind a light-chain comprised of the β2-microglobulin (β2m) due to its association with the α1 and α3 domain and the spatial orientation of these domains [10].

HLA-G expression has been observed in tumour tissues [11]. It has been proposed it functions as an immune checkpoint in cancer as one of the ways in which tumour cells avoid immune cell detection and elimination [12,13]. Indeed, multiple in vitro studies demonstrated HLA-G-expressing tumour cell lines were less susceptible to immune cell-mediated cytotoxicity, as reviewed by Lin et al. [11]. Additionally, it has been frequently reported that HLA-G expression was significantly associated with poor clinical outcome in cancer patients, as reviewed by Carosella et al. [14]. Consequently, HLA-G has been proposed as a candidate target for immune checkpoint inhibitor (ICI) tumour therapy by various studies [14,15,16].

The aim of this review was to uncover to what extent it is justified that HLA-G expression is considered as a target for ICI therapy in carcinomas. To this end, a comprehensive and objective overview concerning the association between HLA-G expression in carcinomas and clinical outcome of patients is provided. This includes the methods and types of analyses that were used to determine tumour HLA-G expression and its clinical impact. Several measures were taken to be able to better compare the various published studies among each other. Included studies were restricted to carcinomas and categorised by anatomical site to gain uniformity. Opposed to already existing HLA-G-related reviews, both statistically significant as well as non-significant analyses concerning the association between tumour HLA-G expression and clinical patient outcome were included to obtain the most complete picture on this association as possible. Additionally, the used HLA-G-detecting mAbs and HLA-G quantification methods were considered when interpreting the results from the different studies. Lastly, not only clinical outcome, but all statistically significant clinicopathological patient and tumour characteristics associated with HLA-G expression were included. In this way, potential patterns between HLA-G expression and tumour development in carcinoma patients were uncovered.

## 2. Association between HLA-G Expression and Clinicopathological Parameters in Carcinoma Patients

### 2.1. Breast Carcinoma

Five studies reported on the association between HLA-G tumour expression and clinical outcome of breast carcinoma patients (Table 1) [17,18,19,20,21]. The first thing that was noticeable was the lack of uniformity in HLA-G quantification methods between these studies [17,18,19,20,21]. It is most likely that the differences in methodologies, as discussed below, have contributed to the wide range in percentage of reported HLA-G-positive tumour samples, ranging from 24 to 62% [18,21]. 

The largest study in sample size was performed by Engels et al. (Table 1) [21]. The main emphasis of this study was on combining multiple markers involved in immune surveillance and tumour immune escape. Tumour immune scores (IS) were generated through a combination of HLA-G, HLA-E, and classical HLA-class I tumour expression and numbers of tumour-infiltrating immune cells expressing FoxP3. Patients were subsequently allocated to low, intermediate or high tumour IS groups. Cox proportional hazard analyses showed significantly increased hazard ratios regarding overall survival (OS), recurrence-free period (RFP) and cancer-specific survival (CSS) in the patient group with low IS opposed to the high IS patient group. Thus, patients with a low tumour IS score showed poor clinical outcome, as expected. A Cox univariate analysis showed that there was no statistically significant association between HLA-G expression and the clinical outcome of patients [21]. HLA-expression only associated with poor clinical patient outcome as one of the parameters that comprised the IS-low group.

By way of comparison, the second largest study by De Kruijf et al. applied similar HLA-G quantification methods as Engels et al. [20,21]. Yet, considerably higher percentages of HLA-G-positive samples (40 vs. 24%) were reported (Table 1) [20,21]. Kaplan–Meier analyses did not reveal statistically significant associations between HLA-G expression and clinical outcome of breast carcinoma patients. However, after stratification of the patient group for loss of HLA-class I expression, HLA-G expression was significantly associated with shorter RFP compared to patients without HLA-G-expressing tumour cells. Thus, both studies demonstrated that tumour HLA-G expression as a single parameter was not significantly associated with poor prognosis in breast carcinoma patients [20,21]. However, by combining tumour HLA-G expression with other factors such as classical HLA-class I tumour status, significantly poor clinical patient outcome was demonstrated [20,21]. Interestingly, in the study by De Kruijf et al., tumour HLA-G expression was positively correlated with tumour HLA-class I expression (Table 1) [20].

Kaplan–Meier analyses performed by Ishibashi et al., Ramos et al. and He et al. revealed significant associations between tumour HLA-G expression and poor clinical outcome of breast carcinoma patients, while all studies used different HLA-G-detecting mAbs and quantification methods (Table 1) [17,18,19]. This significance was maintained in multivariate analyses [18,19]. A notable difference between the studies of Ishibashi et al. and He et al. and those of Engels et al. and De Kruijf et al. was the patient groups that were compared with each other in the survival analyses. Ishibashi et al. and He et al. grouped patients with tumour samples showing absence of HLA-G staining together with samples showing weak staining. This patient group was subsequently compared to patients with moderate to high HLA-G staining in the survival analyses [17,19]. In contrast, Engels et al. and De Kruijf et al. compared patients without HLA-G staining with those showing any staining [20,21]. As a result, the choice for the cut-off value at which patients were considered to have HLA-G-positive tumours, and thus the choice for which patient groups were compared with each other, was different between these studies [17,19,20,21]. Despite the difference in when a tumour sample was considered HLA-G-positive, the percentages of HLA-G-positive tumour samples did not vary as much as one would expect between these studies (Table 1) [17,19,20,21].

These studies, showing statistically significant associations between HLA-G expression as a single parameter and statistically poor clinical outcome of patients, also revealed correlations between HLA-G expression and clinicopathological tumour characteristics associated with worse clinical prognosis [17,18,19]. This includes estrogen receptor (ER) and progesterone receptor (PR) overexpression, advanced disease stage and increased number of lymph node metastases (LNM) (Table 1) [18,19].

In summary, HLA-G quantification methods varied between the studies. This may partly explain variability in range of percentage reported HLA-G-positive tumour samples. Despite these differences, all studies demonstrated significantly poor clinical outcome of patients associated with the highest level of tumour HLA-G expression, either as a single parameter or in conjunction with other immune parameters [17,18,19,20,21]. Additionally, clinicopathological tumour characteristics indicative for disease progression were associated with level of HLA-G expression. This shows, HLA-G associates in many respects with poor clinical outcome in breast carcinomas.

### 2.2. Cervical Carcinoma

To the best of our knowledge, only two studies investigated the correlation between HLA-G expression and clinical outcome of cervical carcinoma patients [22,23]. Significantly poor clinical outcome associated with HLA-G expression was only observed in patients with squamous cell cervical carcinomas with loss of classical HLA-class I molecules in the study by Ferns et al. (Table 2) [22]. Therefore, it seems that the combination of HLA-G upregulation with classical HLA-class I downregulation or loss is indicative for poor clinical outcome. Similar results were found in breast carcinoma patients, as described above [20,21]. Interestingly, an inverse relationship between HLA-G and HLA-class I tumour expression was demonstrated by Rodriguez et al. [23]. Unfortunately, no relation between concomitant tumour HLA-G expression and HLA-class I downregulation and clinical outcome of patients was investigated in this study [23].

Thus, the preliminary available data suggest that tumour HLA-G expression as a single parameter is not associated with poor clinical outcome of cervical carcinoma patients, only in combination with HLA-class I downregulation [22,23].

### 2.3. Colorectal Carcinoma

In total, eight studies described the association between HLA-G expression and clinical outcome in colorectal carcinoma (CRC) patients [24,25,26,27,28,29,30,31]. What stood out during the evaluation of these studies was the considerable discrepancy in HLA-G quantification methods and percentage of HLA-G-positive tumour samples. More importantly, also the association between HLA-G expression and clinical outcome of CRC patients was shown to diverge between the studies. Therefore, we critically assessed whether the use of certain HLA-G quantification methods and the percentage of HLA-G-positive tumours might have had an influence on the eventual clinical outcome of CRC patients.

Ye et al., Cai et al. and Guo et al. showed significant associations between tumour HLA-G expression and poor clinical patient outcome (Table 3) [24,25,29]. The number of patients included in the survival analyses in these studies was relatively low compared to the other CRC studies. All three studies used different HLA-G-detecting mAbs and quantification methods, but no major deviations in the percentage of HLA-G-positive tumour samples were observed, ranging from 59–71% (Table 3) [25,29]. Contrarily, Lin et al. and Zeestraten et al. did not show statistically significant associations between tumour HLA-G expression and clinical patient outcome, despite considerably diverging percentages of HLA-G-expressing tumour samples (71 vs. 20%, respectively) [26,30].

Interestingly, in studies where similar HLA-G quantification methods were used, considerable differences in the percentage of HLA-G-positive tumour samples and clinical patient outcome were observed. For instance, Guo et al. and Zeestraten et al. applied similar HLA-G quantification methods and both used HLA-G-detecting mAbs that should recognise all HLA-G isoforms (MEM-G/2 and 4H84, respectively) [29,30]. Yet, Guo et al. reported that 71% of tumour samples was HLA-G positive, while only 20% was deemed HLA-G positive in the study by Zeestraten et al. [29,30]. Additionally, no agreements concerning the association between HLA-G expression and clinical patient outcome was reached between these two studies [29,30].

Wide discordances concerning the association between tumour HLA-G expression and clinical patient outcome were also reported independent of the percentage of HLA-G-positive tumour samples and used methods. As such, both Guo et al. and Lin et al. reported that 71% of the CRC patients had HLA-G-positive tumours [26,29]. Yet, Guo et al. did demonstrate poor clinical outcome of patients associated with HLA-G expression, while Lin et al. reported no relationship between these two parameters [26,29].

The general discord concerning the association between tumour HLA-G expression and clinical outcome of CRC patients is emphasized in the study by Reimers et al. [31]. They reported that patients with tumours showing strong staining for HLA-G expression had significantly better clinical outcome (Table 3). Interestingly, also a positive correlation between tumour samples that stained strongly for HLA-G and HLA-class I expression was observed.

Zhang et al. and Kirana et al. reported mixed results within their own study with respect to HLA-G-related clinical outcome of patients [27,28]. Statistically significant, as well as nonsignificant associations between tumour HLA-G expression and clinical outcome of patients were observed (Table 3). Especially the way in which the mixed results were established in the study by Zhang et al. were of particular interest [27]. Here, two separate cut-off values were chosen at which IHC tumour samples were deemed HLA-G positive (Table 3). The lower threshold value assumed that IHC tumour samples were HLA-G positive when ≥5% of the tumour cells stained for HLA-G expression. The higher threshold value was set at ≥55%. With the threshold set at ≥5%, no significant difference in OS time was observed. However, when the threshold was set at ≥55%, significantly shorter OS time in association with HLA-G expression was found. This association even remained statistically significant in multivariate analysis. Thus, statistical significance regarding the association between HLA-G expression and shorter OS time was reached by raising the threshold value for HLA-G detection.

Regarding the clinicopathological factors, studies showing significant association between tumour HLA-G expression and poor patient survival, also observed positive correlations between HLA-G expression and advanced TNM stages [24,25]. Conversely, Lin et al. did not find an association between tumour HLA-G expression and clinical outcome of patients and even reported an inverse relationship between tumour HLA-G expression and TNM stage [26]. On top of that, Reimers et al. described that advanced TNM stages were predominantly found in patients with tumours that stained weakly for HLA-G expression and that HLA-G upregulation rather led to prolonged instead of shortened survival times [31].

In conclusion, several studies demonstrated that tumour HLA-G expression was associated with poor clinal outcome of CRC patients, while other studies that yield more statistical power due to larger sample size did not observe these associations [24,25,26,29,30]. Two other studies showed mixed results concerning the HLA-G-related clinical outcome of patients [27,28]. Reimers et al. even reported prolonged DFS time in CRC patients with upregulated HLA-G expression [31]. Thus, the role of HLA-G in CRC patients is currently not clear and requires further attention.

### 2.4. Esophageal Carcinoma

Three studies reported on the correlation between tumour HLA-G expression and clinical outcome of esophageal squamous cell carcinoma (ESCC) patients (Table 4) [32,33,34]. All three studies showed significantly poor OS associated with tumour HLA-G expression [32,33,34]. Yie et al. and Lin et al. demonstrated that the correlation between tumour HLA-G expression and poor clinical outcome maintained its statistical significance in multivariate analysis, despite the fact that the number of included patients in the survival analyses was relatively low (70 and 40 patients, respectively) [32,33]. Furthermore, associations between tumour HLA-G expression and indicators of advanced cancer stages, such as advanced TNM and increased LNM, were observed in all included studies [32,33,34].

### 2.5. Gastric Carcinoma

Six studies investigated the association between tumour HLA-G expression and clinical outcome of gastric carcinoma (GC) patients (Table 5) [35,36,37,38,39,40]. Except for one study, the verdict on the association between HLA-G expression and clinical outcome of GC patients is relatively unanimous in comparison to other carcinoma types. The majority of the studies concluded that HLA-G expression correlated with significantly shorter patient survival times [35,37,38,39,40]. This significance was often maintained in multivariate analyses [35,37,38,40]. A disclaimer has to be added stating that all studies presented procedural differences, such as different HLA-G-detecting mAbs and HLA-G quantification methods, of which the consequences on the eventual study outcomes cannot be accounted for. In contrast to all other GC studies, Ishigami et al. reported significantly prolonged OS time of GC patients associated with HLA-G expression [36]. No particular factor could be pinpointed that would explain this inverted outcome opposed to the other studies.

Studies showing significant associations between HLA-G and poor clinical patient outcome also found positive correlations between HLA-G expression and tumour grade, stage, depth and LNM (Table 5) [35,38,39,40]. These clinicopathological factors are indicative for increased tumour burden. In contrast to all other studies, Ishigami et al. reported inverse correlations of tumour HLA-G expression and these parameters [36].

Noteworthy, it has been repeatedly demonstrated that tumour HLA-G expression correlated with immune-related parameters in GC patients. Tumour-infiltrating NK and CD8+ T cells were negatively associated with HLA-G expression, whereas infiltrating Tregs were positively correlated with HLA-G expression (Table 5) [35,38,39,40]. This may indicate a potential functional role for HLA-G in modulating tumour-immune responses, potentially culminating in disease progression of patients with GC.

In conclusion, HLA-G expression was associated with significantly poor clinical outcome in GC patients and also correlated with parameters that reflect low immunogenicity/immunosuppression of tumours. Therefore, HLA-G expression may serve as an immune checkpoint molecule in GC.

### 2.6. Hepatocellular Carcinoma

To the best of our knowledge, only two papers have been published regarding the correlation between HLA-G expression and clinical outcome in hepatocellular carcinoma (HCC) patients [41,42].

Wang et al. used Western Blot (WB) analysis with MEM-G/1 mAbs to quantify the percentage of HLA-G-positive HCC samples (Table 6) [41]. Significantly shorter OS and RFP were observed in patients with HLA-G expression when assessed as a single parameter [41]. However, relatively low patient numbers (*n* = 36) were used in the survival analysis and no multivariate analysis was performed [41].

The study by Cai et al. also showed that HLA-G expression was associated with significantly shorter OS time in HCC patients (Table 6) [42]. Further analyses revealed that HLA-G expression was correlated with poor clinical outcome of patients in predominantly the early HCC stages. The data concerning the association between HLA-G expression and clinical outcome in intermediate and advanced stages were not presented, but were said to be “not inspiring’’ [42]. These results suggest that tumour HLA-G expression plays a potential role in early HCC only. Furthermore, tumour HLA-G expression was positively correlated with the Treg/CD8+ ratio, which signifies a potential functional role of HLA-G in modulating the immune-tumour response [42].

Concluding, only a limited number of studies was available that addressed the association between HLA-G expression and clinical outcome in HCC patients. Therefore, it is advised to take caution when interpreting these results. Based on the currently available data, HLA-G expression seems to be significantly correlated with poor clinical patient outcome [41,42]. It is strongly advised to perform additional research to be able to be more conclusive on this topic.

### 2.7. Lung Carcinoma

Four studies reported on the association between tumour HLA-G expression and clinical outcome of non-small-cell lung carcinoma (NSCLC) patients (Table 7) [43,44,45,46]. In general, these studies are characterised by relatively low patient numbers in the survival analyses. Additionally, particular attention is given to soluble HLA-G (sHLA-G) isoforms in NSCLC patients compared to other cancer types. For instance, the largest included study by Yan et al. investigated sHLA-G expression in tumour lesions of 123 patients [44]. The mAb 5A6G7 was used as HLA-G-detecting mAb that recognises an epitope on intron 4 only present on the soluble isoforms HLA-G5 and -G6 [44]. As a result, all membrane-bound HLA-G isoforms remain undetected, which may have led to a high false-negative rate in the search for total HLA-G expression. Nevertheless, no relation between HLA-G5 and -G6 expression and OS time was found [44]. Interestingly, Lin et al. investigated both tumour and serum HLA-G expression [43]. Serum sHLA-G expression was determined with a sHLA-G-specific ELISA-kit. It was shown that lesion HLA-G expression did not associate with clinical outcome of patients, but sHLA-G expression in the serum did correlate with significantly shorter OS time [43]. To complicate matters even more, it was demonstrated that the serum sHLA-G levels did not correlate with tumour HLA-G expression within individual patients [43].

Regarding the other studies, both Yie et al. and Zhang et al. observed significant associations between tumour HLA-G expression and shorter OS time [45,46]. Both studies also observed significant associations between HLA-G expression and clinical parameters related to increased tumour burden, such as increased LNM and advanced disease stages [45,46].

In conclusion, the results on the association between HLA-G expression and clinical outcome of NSCLC patients are equivocal between the various studies. Therefore, more studies with more patients included in the survival analysis are needed before a proper verdict on the association between HLA-G expression and clinical outcome in NSCLC patients can be provided.

### 2.8. Oral Carcinoma

Three studies reported on the association between HLA-G expression and clinical outcome of oral carcinoma patients [47,48,49]. These studies were characterised by low patient numbers, ranging from 33 to 60 patients (Table 8) [47,48,49].

Only a significant correlation between HLA-G expression in the tumour parenchyma and shorter OS time was observed through Spearman’s correlation coefficient analysis by Imani et al. [48]. Remarkably, HLA-G expression was also observed in the tumour stroma. It remains unclear which cells expressed HLA-G in the stroma from the available data, e.g., fibroblasts or immune cells. No significant correlations were found between HLA-G expression in the tumour stroma and OS time within the same study. Additionally, positive correlations between tumour HLA-G expression and clinicopathological parameters such as tumour stage, LNM and distant metastasis were only observed in the tumour parenchyma and not in the stroma. Furthermore, Kaplan–Meier analysis did not reveal any statistically significant differences in OS time associated with tumour HLA-G expression, although it was unclear which patient groups were being compared with each other [48].

Regarding the studies by Gonçalves et al. and Mosconi et al., both concluded that HLA-G expression was not associated with significantly shorter OS time [47,49]. Nevertheless, increased tumour depth and advanced histological tumour grade were still shown to be associated with tumour HLA-G expression [47,49].

Concerning the used methods, Gonçalves et al. used IHC analysis with MEM-G/2 as HLA-G-detecting mAb. [49]. IHC samples were scored using an Immune Reactivity Score (IRS) (Table 8) [49]. As such, 50% of the tumour samples expressed high levels of HLA-G (IRS ≥ 2) [47]. By comparison, Imani et al. used similar HLA-G quantification methods, but used 4H84 instead of MEM-G/2 as HLA-G-detecting mAb [48]. They observed that 33% of the tumour samples expressed high levels of HLA-G [48]. Thus, both studies used similar HLA-G quantification methods and used HLA-G-detecting mAbs that should recognise all HLA-G isoforms. Yet, considerable differences in tumour samples having high HLA-G expression were declared between these two studies (33% with 4H84 vs. 50% with MEM-G/2) [47,48].

Concluding, tumour HLA-G expression does not seem to significantly associate with OS time in oral carcinoma patients [47,48,49]. Yet, several clinicopathological factors associated with increased tumour burden were associated with HLA-G expression [47,48,49]. Overall, study sample sizes should be upscaled before any definitive conclusions concerning the association between HLA-G expression and clinical outcome of oral carcinoma patients can be drawn.

### 2.9. Ovarian Carcinoma

Five studies investigated the association between HLA-G expression and clinical outcome of ovarian carcinoma patients (Table 9) [50,51,52,53,54]. The majority of these studies included a random cohort of ovarian carcinoma patients, while Rutten et al. and Andersson et al. focussed on advanced stage ovarian carcinoma patients [50,51,52,53,54].

Jung et al. reported that tumour HLA-G expression was associated with significantly poor clinical patient outcome when using IHC analysis to determine the percentage of HLA-G-positive tumour samples [52]. However, only OS and not progression-free survival (PFS) was significantly associated with HLA-G expression. Additionally, no statistically significant associations between tumour HLA-G expression and clinical patient outcome were found after stratification for specific cancer stages, in multivariate analysis or when using WB analysis to determine the percentage of HLA-G-positive tumour samples (Table 9) [52]. Andersson et al. also reported significantly poor OS time associated with HLA-G expression, but only in a specific patient subgroup (Table 9) [50]. The unstratified Kaplan–Meier analyses did not reveal differences in patient survival times associated with HLA-G expression [50].

The study by Rutten et al. focussed on tumour HLA-G expression in high-grade ovarian carcinoma patients (Table 9) [53]. Significantly improved clinical outcome of patients associated with HLA-G expression was observed using Kaplan–Meier analysis. Tumour samples were also stained for HLA-class I expression with HCA2 mAbs, which preferentially bind to HLA-A heavy chains [55]. Interestingly, after stratification for patients with downregulated HLA-A expression, HLA-G expression was no longer significantly associated with improved clinical patient outcome (Table 9) [53]. Thus, the beneficial effects that HLA-G expression might have had in ovarian carcinoma patients possibly depended on HLA-A co-expression.

Babay et al. and Zhang et al. did not observe significant correlations between tumour HLA-G expression and clinical outcome of ovarian carcinoma patients [51,54]. It is noteworthy that, in the study by Zhang et al., HLA-G expression had a tendency towards having a beneficial effect on clinical outcome (Table 9) [54].

Regarding the clinicopathological factors, Jung et al. reported that tumour HLA-G expression was correlated with advanced cancer stage [52]. All other studies did not demonstrate correlations between tumour HLA-G expression and factors indicative for increased tumour burden in ovarian carcinoma patients (Table 9) [50,51,53,54].

The percentage of HLA-G-positive tumour samples differed considerably between the included studies, ranging from 20 to 82% (Table 9) [50,54]. The disparities between the reported percentages of HLA-G-positive tumour samples cannot be attributed to significant differences in HLA-G quantification methods. For example, in the studies by Babay et al. and Andersson et al., tumour samples were considered HLA-G positive when ≥1% of the tumour cells was stained for HLA-G [50,51]. Both studies used HLA-G-detecting mAbs that recognise all HLA-G isoforms, namely MEM-G/1 and 4H84 mAbs. Yet, considerably higher percentages of HLA-G-positive tumour samples were reported by Babay et al. using 4H84 mAbs (72% vs. 20%) [50,51]. These variations could arise from methodological differences in IHC staining and blocking processes between these studies. However, the true reasons remain elusive, as these processes were barely described.

In conclusion, the majority of the included studies demonstrated either no significant associations or even beneficial associations between HLA-G expression as a single parameter and clinical outcome of ovarian carcinoma patients [50,51,53,54].

### 2.10. Pancreatic Carcinoma

Four studies investigated the association between tumour HLA-G expression and clinical outcome of pancreatic carcinoma patients (Table 10) [56,57,58,59].

Sideras et al. included the largest patient cohort consisting of both pancreatic and ampullary carcinoma patients (Table 10) [57]. Significantly prolonged OS and DFS time were observed associated with tumour HLA-G expression. However, statistical significance was abolished after stratification for tumour site (pancreas vs. ampulla) and in multivariate analysis [57].

Conversely, Zhou et al., Xu et al. and Hiraoka et al. concluded that HLA-G expression was associated with poor clinical outcome of pancreatic carcinoma patients (Table 10) [56,58,59]. Interestingly, Zhou et al. demonstrated that HLA-G expression was correlated to decreased tumour infiltrating lymphocytes (TILs), possibly indicating a functional role of HLA-G expression in influencing the immune-tumour response in pancreatic carcinoma [59]. Additionally, it has to be mentioned that, unfortunately, only the abstract of the article by Xu et al. was available [58].

The HLA-G quantification methods varied considerably between the included studies [56,57,58,59]. Zhou et al. used unconventionally high cut-off values at which tumour samples were deemed HLA-G positive (≥ 75%) relative to what is more commonly observed [59]. They also did not mention the used HLA-G-detecting mAb which greatly hampers the opportunity to fairly assess their HLA-G quantification methods [59].

Despite the differences in HLA-G quantification methods between the studies by Zhou et al. and Sideras et al., both constructed tissue microarrays and reported nearly identical percentages of HLA-G-positive tumour samples (14 vs. 15%, respectively) [57,59]. Nonetheless, their verdict on the association between tumour HLA-G expression and clinical outcome are contradictory to one another, as described above and shown in Table 10. Thus, there is no apparent link between the percentage of HLA-G-positive tumour samples and the subsequent verdict on the association between HLA-G expression and clinical outcome of pancreatic carcinoma patients.

In conclusion, the majority of studies concluded that HLA-G expression was associated with poor clinical patient outcome [56,58,59]. However, the largest study performed by Sideras et al. concluded that HLA-G expression as single parameter was correlated with improved clinical patient outcome [57]. Thus, considerable contradictions were observed concerning the association between HLA-G expression and clinical outcome of pancreatic carcinoma patients between different studies. Therefore, further research on the association between HLA-G expression and clinical outcome of pancreatic carcinoma patients is warranted. When doing so, it is strongly recommended to apply similar HLA-G quantification methods in order to be able to better compare the studies among each other. Until then, no conclusive verdict can be given on the role of HLA-G in pancreatic carcinoma.

### 2.11. Residual Carcinoma Types

HLA-G-related studies have also been performed in bladder, prostate and skin carcinoma. However, none of these studies performed analyses concerning the association between tumour HLA-G expression and clinical patient outcomes. A single article has been published on the correlation between HLA-G expression and clinical patient outcome in endometrial, lip, renal and thyroid carcinoma patients [60,61,62,63]. As a result, no comparisons between studies within these carcinoma types could be made. However, in order to provide a complete overview of all studies discussing the association between HLA-G expression and clinical outcome of patients, the studies on these cancer types are summarised in Table 11.

Bijen et al. investigated the association between tumour HLA-G expression and clinical outcome of endometrial carcinoma patients [60]. In total, 209 out of 525 patients (40%) had either low or strong tumour HLA-G expression (Table 11). However, the number of patients with HLA-G-positive samples included in the survival analysis remains unclear. Nonetheless, no survival-related factors were associated with tumour HLA-G expression in endometrial carcinoma patients [60]. Tumour HLA-G expression also showed an association with HLA-class I overexpression [60].

Lopes et al. reported there was no correlation between tumour HLA-G expression and clinical outcome in lip squamous cell carcinoma (LSCC) patients [61]. They also mentioned that HLA-G expression was not related to the number of cells expressing CD8 and granzyme B in the tumour lesion. These molecules are known markers for cytotoxic T lymphocytes and NK cells. Furthermore, all LSCC samples were HLA-G positive, whilst the distribution of HLA-G-positive samples amongst the various staining groups was not provided (Table 11) [61].

De Figueiredo-Feitosa et al. did not observe significantly shorter DFS time in papillary thyroid carcinoma patients related to tumour HLA-G expression (Table 11) [62]. HLA-G staining was done using 5A6G7 mAbs that only recognise sHLA-G isoforms. Despite that only a particular subset of HLA-G isoforms was stained for, all patients with papillary as well as follicular thyroid carcinoma were deemed HLA-G positive [62].

In the study by Jasinski-Bergner et al., renal cell carcinoma (RCC) samples were quantified for tumour HLA-G expression through IRSs. The precise distribution of HLA-G-positive samples across the various staining intensities was not mentioned and it was unclear when tumour samples were considered HLA-G positive. Nonetheless, no association between HLA-G expression and clinical outcome of RCC patients was observed [63]. Furthermore, additional IHC staining was performed on a selected group of 36 HLA-G-positive and 36 HLA-G-negative RCC samples [63]. In this selected group of 72 samples, numbers of CD3+ and CD8+ T cells were significantly increased in the HLA-G-positive group. Interestingly, FoxP3-positive Treg cells, CD4+ T cells and CD56+ NK cells were not significantly associated with tumour HLA-G expression. Furthermore, Friedrich et al. also performed HLA-G-related survival analyses in RCC patients, but used the same patient cohort as Jasinski-Bergner et al. [64]. Therefore, this study is not included as a separate article in this review. Friedrich et al. mainly focussed on the relation between tumour HLA-G and cAMP response element binding protein (CREB) expression [64]. Unsurprisingly, similar to the study by Jasinski-Bergner et al., the association between tumour HLA-G expression and survival were not significant.

Concluding, it remains up to debate what the true impact is of HLA-G expression on clinical patient outcome in these particular carcinoma types due to the scarcity of publications. Additional studies should be performed to validate these preliminary findings and to conclude whether HLA-G plays a role in these carcinoma types.

## 3. Discussion

The aim of this review was to obtain a complete and objective view on the current status of the relationship between tumour HLA-G expression and clinicopathological parameters, including clinical outcome, in carcinoma patients. The main observation was that HLA-G-related research is characterised by heterogeneity. Not only in used HLA-G-detecting mAbs, quantification methods and stratification variables, but also in conclusions concerning clinical outcome of carcinoma patients. Tumour HLA-G expression was significantly associated with poor clinical patient outcome in the majority of studies regarding breast, gastric, hepatocellular and esophageal carcinomas. It is implied that in the carcinoma types where HLA-G expression was associated with poor clinical patient outcome, HLA-G expression plays a functional role in disease progression and, therefore, is a candidate immune checkpoint molecule. Cervical, colorectal, lung, and pancreatic carcinoma patients presented discrepant and, therefore, inconclusive results regarding the association between tumour HLA-G expression and clinical outcome. No associations between tumour HLA-G expression and clinical outcome were observed in patients with oral and ovarian carcinomas. This suggests that HLA-G expression plays different roles in distinct carcinoma types. However, it should be noted that considerable methodological variations brought uncertainty into the interpretation of the reported results. Hence, the necessity for standardised operating procedures in future HLA-G-related research has become all the more evident. In our opinion, it can therefore not be concluded with certainty that HLA-G is an immune checkpoint in all these carcinoma types.

The majority of included studies used 4H84 as HLA-G-detecting mAb in IHC analysis to quantify the percentage of HLA-G-positive carcinoma samples. However, 4H84 mAbs bring forth their own particular set of drawbacks. First, 4H84 mAbs have shown non-specific binding and cross-reactivity with classical HLA-class I molecules [65,66,67,68]. As a result, overestimation of HLA-G expression in tumour lesions is a realistic possibility, which in turn influences the HLA-G-related clinical outcome analyses in carcinoma patients. Secondly, 4H84 mAbs recognise an epitope on the α1 domain present on all HLA-G isoforms. On the one hand, this provides the most complete picture of the total HLA-G expression in the tumour lesion. On the other hand, nothing on the expression of different HLA-G isoforms can be concluded. It has been demonstrated that the type of HLA-G isoform and its configuration are important for the interaction HLA-G can engage with specific HLA-G receptors presented on immune cells [69,70]. In theory, HLA-G isoform expression patterns may vary among and within carcinoma types. The exclusive expression of certain HLA-G isoforms by particular carcinoma types also provides an explanation for why HLA-G expression seems to play different roles in distinct carcinoma types. However, no discrimination can currently be made between the different HLA-G isoforms in the tumour lesions when 4H84 (or any mAb that recognises all HLA-G isoforms for that matter) is used as HLA-G-detecting mAb. Therefore, until it becomes possible to make a distinction between the different HLA-G isoforms and determine the proportions wherein distinct HLA-G isoforms are presented within the tumour lesion, it remains uncertain what the exact role of specific HLA-G isoforms is in different tumour types [71]. A solution to these issues would be the development of mAbs that exclusively stain the distinct HLA-G isoforms, but that is challenging.

Other approaches to measure HLA-G in tumour lesions could include using HLA-G mRNA expression levels. However, it has been shown in CRC samples that HLA-G mRNA expression levels does not necessarily translate to HLA-G protein expression [71,72]. This is because HLA-G mRNA is under strict post-translational control by HLA-G mRNA-specific miRNA’s [63,73,74,75]. Therefore, more upstream indications of presence of HLA-G in tumour lesions (e.g., through mRNA expression) is not a fair reflection of the actual expression of HLA-G on protein level.

Special consideration went into whether the used methods to determine tumour HLA-G expression were indicative for the association between tumour HLA-G expression and clinical patient outcome. Studies with comparable HLA-G quantification methods differed considerably in the percentage of HLA-G-positive tumour samples (Engels et al. vs. De Kruijf et al. in breast carcinoma [20,21]; Lin et al. and Guo et al. vs. Zeestraten et al. in CRC [26,29,30]; Lin et al. vs. Yie et al. in ESCC [32,33]; Du et al. vs. Yie et al. in GC [35,40]; Lin et al. vs. Yie et al. in NSCLC [43,45]; Gonçalves et al. vs. Imani et al. in oral carcinoma [47,48]; Babay et al. vs. Andersson et al. in ovarian carcinoma [50,51]). In part, this can be ascribed to the use of different HLA-G-detecting mAbs. 4H84, MEM-G/1, MEM-G/2 and HGY are all able to recognise all HLA-G isoforms. Yet, Swets et al. showed that variations in HLA-G binding patterns exists between 4H84 mAbs and MEM-G/1 and MEM-G/2 in sequential CRC tissue sections [68]. In general, studies staining for HLA-G with 4H84 mAbs resulted in higher percentages of HLA-G-positive tumour samples than with MEM-G/1 or MEM-G/2 mAbs, while using similar HLA-G quantification methods (Guo et al. vs. Zeestraten et al. in CRC [29,30]; Gonçalves et al. vs. Imani et al. in oral carcinoma [47,48]; Babay et al. vs. Andersson et al. in ovarian carcinoma [50,51]). Additionally, a particular research group used HGY, a non-commercially available HLA-G-detecting mAb [19,24,33,40,45]. Although similar specificity and affinity as 4H84 mAbs was claimed, the HLA-G-positive sample rate showed a tendency to be higher compared to 4H84 mAbs with similar HLA-G quantification methods (Ye et al. vs. Zeestraten et al. in CRC [24,30]; Lin et al. vs. Yie et al. in ESCC [32,33]; Du et al. vs. Yie et al. in GC [35,40]; Lin et al. vs. Yie et al. in NSCLC [43,45]). Remarkably, all studies using HGY as HLA-G-detecting mAbs concluded that tumour HLA-G expression was significantly associated with shorter OS in breast, colorectal, esophageal, gastric and lung carcinoma patients. No other apparent links could be distinguished between the percentage of HLA-G-positive tumour samples and clinical outcome of patients. Even in studies where the reported percentage of HLA-G-positive tumour samples was comparable, regardless of the used methodology, no consensus concerning the association between tumour HLA-G expression and clinical outcome of patients was reached (Lin et al. and Zhang et al. vs. Guo et al. in CRC [26,27,29]; Du et al. vs. Ishigami et al. in GC [35,36]; Sideras et al. vs. Zhou et al. in pancreatic carcinoma [57,59]). Additionally, no agreements on HLA-G expression levels could be observed between IHC and WB analysis with 4H84, MEM-G/9 and 5A6G7 in CRC [76]. Taken together, it seems that there is no correlation between the used method to quantify the HLA-G expression, the percentage of reported HLA-G-positive tumours and the eventual clinical outcome of patients. All this is indicative for the general sensitivity for subjectivity in HLA-G-related research.

Although the effect of methodological differences on the eventual results remains largely unknown, it did become clear that the choice for a particular cut-off value at which IHC samples were deemed HLA-G-positive had a significant impact on study results, as demonstrated by Zhang et al. [27]. Herein, statistical significance for clinical outcome of CRC patients was reached by raising the cut-off value from 5 to 55% of cells stained. Furthermore, the study by Lin et al. in ESCC demonstrated that the hazard ratio (HR) was higher for patients with tumour samples showing >50% of cells stained for HLA-G vs. no staining (HR = 3.02, *p*-value ≤ 0.001) opposed to patients with 1–50% staining vs. no staining (HR = 2.02, *p*-value = 0.01) [32]. These two examples indicate that in particular high HLA-G tumour staining associates with clinical parameters.

It is important to keep in mind that in cancer biology the interaction between tumour cells and surveilling immune cells is a complex multifactorial process. Many different activating and inhibiting molecules, on many different cell types, are involved in tumour-immune interactions. Therefore, it is not easy to reduce such clinical patient outcome solely to a component like HLA-G. Therefore, in order to gain more understanding of the function of HLA-G expression in cancer, we recommend for future HLA-G-related research to shift the focus from investigating the singular association of HLA-G on clinical outcome to beholding HLA-G as one of many factors that is involved in tumour progression. For instance, several studies demonstrated that the combination of HLA-class I downregulation and HLA-G expression in tumours was correlated with poor clinical patient outcome in multiple carcinoma types [20,21,22,53]. The opposite was also true. Studies showing simultaneous expression of tumour HLA-class I and HLA-G molecules did not reveal significantly poor clinical patient outcome [20,31,60]. This apparent interaction between tumour HLA-class I expression, HLA-G expression and its association with clinical patient outcome may provide an explanation for the biological course of action of tumour immunoediting. According to the immunoediting theory, tumour cells with less HLA-class I molecules are less immunogenic for T cell-mediated immune responses, and, therefore, have a selective advantage for outgrowth. Consequently, tumour cells become targets for NK cells because of the recognition of the ‘missing-self’, represented by the lack of HLA-class I expression [77]. Upregulation of HLA-G may provide tumour cells with protection from NK cell-mediated cytotoxicity as HLA-G inhibits NK cell function [78].

Another factor that should be highlighted in combination with HLA-G expression in tumours is the presence of infiltrating immune cells [79]. The HLA-G receptors ILT2, ILT4 and KIR2DL4 are primarily expressed on immune cells, as reviewed by Attia et al. [70]. So, in order for HLA-G to exert a functional role in tumour progression, it will be in interaction with immune cells. Especially in GC patients, correlations between tumour HLA-G expression and decreases in number of tumour-infiltrating NK and CD8+ cells and increases in number of Tregs were frequently observed [35,38,39,40]. This observation indicates that tumour HLA-G may be involved in modulating the tumour-immune response, which could result in tumour progression, and therefore assume the role of immune checkpoint molecule in GC patients.

Studies showing significant associations between tumour HLA-G expression and poor clinical patient outcome also found positive correlations between HLA-G expression and clinicopathological parameters associated with increased tumour burden, such as tumour grade, stage, depth and LNM [18,19,24,25,32,33,34,35,38,39,40,45,46,52,58,59]. The opposite was also true. Studies showing inverse relationships between HLA-G expression and clinicopathological parameters associated with increased tumour burden showed prolonged survival or statistically non-significant poor clinical outcome associated with HLA-G expression [26,31,36]. In general, studies presenting no significant association between tumour HLA-G expression and clinical patient outcome were also less likely to report significant correlations between HLA-G expression and parameters indicative for increased tumour load. Thus, it seems that tumour HLA-G expression, clinicopathological parameters indicative for tumour burden, and clinical outcome of patients are interrelated. An important question that arises from these observations is whether the relationship between HLA-G expression and clinical patient outcome is a causal relationship. Or rephrased, has HLA-G an active, functional role in tumour progression which leads to poor clinical patient outcome or is HLA-G expression a by-product of an increased genomic instability in the tumours of patients with poor clinical outcome? It may be possible that tumour HLA-G expression is one of the hallmarks of tumours with more aggressive phenotypes resulting from mutations associated with increased tumour burden. If this would be the case then HLA-G expression does not have a causal, functional role in tumour progression and therefore poor clinical patient outcome, but would rather be the consequence of these tumours possessing an aggressive phenotype (Figure 1).

To be a candidate target for ICI therapy in certain carcinoma types, there should be a clear correlation between tumour HLA-G expression and a poor clinical outcome compared to non-HLA-G-expressing tumours of that carcinoma type. If this association cannot be observed, it is unlikely there is a causal relationship between tumour HLA-G expression and inhibition of an anti-tumour immune response. Although the effects of tumour HLA-G expression on clinical patient outcome remains enigmatic, overall it seems obvious that HLA-G can be expressed by tumour cells, albeit heterogenic within and between tumours [11,80,81]. Moreover, the majority of included studies showed correlations between HLA-G expression and shorter patient survival time, despite these not always reaching statistical significance. Additionally, promising results have been shown in halting the progression and eradication of HLA-G1-expressing tumour cells using CAR-T cells directed specifically against HLA-G in an in vivo mouse model [82]. Based on the current information, breast, gastric, hepatocellular and esophageal carcinoma might qualify for ICI HLA-G therapy, while cervical, colorectal, lung, oral, ovarian and pancreatic carcinoma still does not.

In general, a considerable number of questions remain unanswered concerning HLA-G expression in carcinomas, including: what is the most reliable way to quantify HLA-G expression? Which HLA-G isoforms are being expressed? What is the relation between HLA-G expression and other factors involved in tumour progression such as HLA-class I downregulation and the presence of infiltrating immune cells? Additionally, is the relation between HLA-G expression and poor clinical outcome of patients causal or not? As long as these questions remain largely unanswered, it is premature to conclude that HLA-G expression is an immune checkpoint molecule in carcinomas.

## 4. Conclusions

In summary, HLA-G expression in breast, gastric, hepatocellular and esophageal carcinoma patients was associated with significantly poor clinical outcome. HLA-G might therefore have a functional role in these carcinoma types and thus might act as an immune checkpoint molecule. Cervical, colorectal, lung, oral, ovarian and pancreatic carcinoma patients did not univocally reveal poor clinical outcome associated with HLA-G expression. It therefore remains uncertain what the exact role of HLA-G expression is in these carcinoma types. We conclude that, before any therapeutic application can be applied that targets HLA-G, more information is needed on tumour expression of HLA-G and of its isoforms. In addition, the development of novel and more specific HLA-G-detecting mAbs is emphasised. Additionally, future research should focus on the role of HLA-G in tumour-immune interactions, as this is a multifactorial and complex process. In our opinion, the declaration of HLA-G as novel immune checkpoint molecule in cancer is premature.

## Figures and Tables

**Figure 1 ijms-22-08265-f001:**
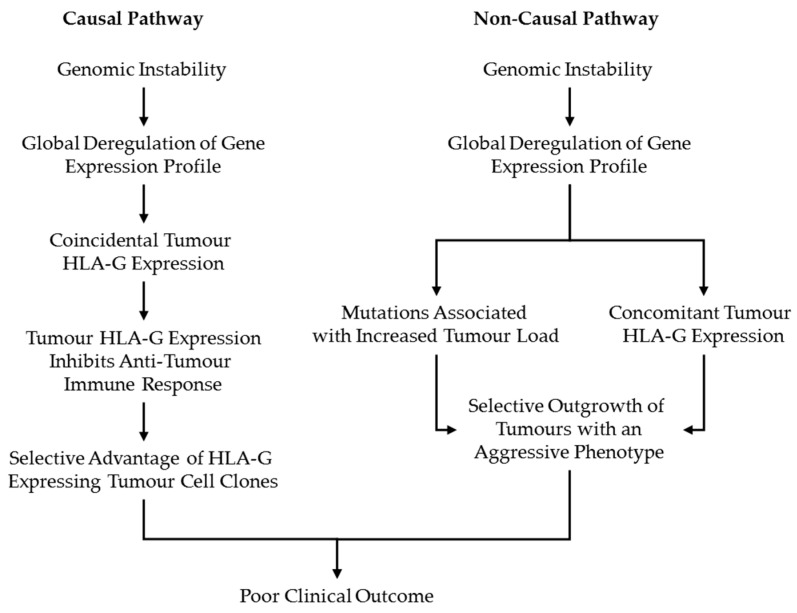
Alternative explanation for the relationship between HLA-G expression and clinical outcome in carcinoma patients. The current perspective in HLA-G-related research on the association between HLA-G expression and clinical outcome of carcinoma patients dictates that HLA-G expression aides tumour cells in the immunoediting process. As a result, HLA-G-expressing tumour cells have a selective advantage over the tumour cell clones that do not express HLA-G by supporting the outgrowth and persistence of HLA-G-positive tumour cell clones. This eventually results in poor clinical patient outcome of patients with HLA-G-expressing tumours (causal pathway). The proposed alternative explanation suggests that poor clinical outcome of carcinoma patients is predominantly the result of that certain tumours possess more aggressive phenotypes, and that these tumours concomitantly express increased levels of HLA-G. In this example, the contribution of tumour HLA-G expression to this aggressive phenotype is unclear. Consequently, the functional role of HLA-G expression as actual mediator of tumour progression and poor clinical outcome in carcinoma patients is questionable (non-causal pathway).

**Table 1 ijms-22-08265-t001:** Studies on the Correlations between Tumour HLA-G Expression and Clinicopathological Factors and Clinical Outcome of Breast Carcinoma Patients.

First Author [Ref.]	mAb and Included Patient Cohort	HLA-G Quantification Method	HLA-G+ Samples (%)	Association with Tumour HLA-G Expression	
				Clinico-Pathological Parameters with *p*-Values ≤ 0.05	Clinical Outcome (*p*-Value)
* Engels [21]	4H84 Post-meno-pausa, hormone receptor positive BC patients	Tumour was considered HLA-G-positive when >1% of tumour cells were stained Tumour immune-susceptibility was expressed in IS and was generated by adding up regression coefficients of HLA-G, HLA-class I, HLA-E and FoxP3 expression, thereby creating three groups; low, intermediate and high IS	Low IS: 817/1636 (50) Intermediate IS: 318/1636 (19) High IS: 501/1636 (31) Total cohort: 484/2042 (24)	Increased tumour grade.	Cox univariate analysis: OS (ns); RFP (ns); CSS (ns). Cox proportional hazard analysis, intermediate vs. high IS: OS, HR = 1.471 (none provided); RFP, HR = 1.539 (none provided); CSS, HR = 2.119 (none provided). Cox proportional hazard analysis, low vs. high IS: Shorter OS, HR = 1.602 (0.002); Shorter RFP, HR = 1.634 (0.002); Shorter CSS, HR = 2.103 (<0.001).
* De Kruijf [20]	4H84 Early BC patients	Tumour was considered HLA-G-positive when >1% of tumour cells were stained	201/501 (40)	HLA-class I expression; Her2 over-expression; Type of received systemic therapy.	KM analysis: OS (ns); RFP (ns). KM analysis, stratified for HLA-class I expression, *n* = 361: OS (ns); RFP (ns). KM analysis, stratified for loss of HLA-class I expression, *n* = 106: OS (ns); Shorter RFP (0.035).
Ishibashi [17]	4H84 BC patients, random cohort	Low (absent (-) or weak (+)) staining vs. high staining (moderate (++) or strong (+++))	+: 58/102 (57) ++: 32/102 (31) +++: 6/102 (6) High staining: 38/102 (37)	Tumour ER down-regulation; Tumour PR down-regulation.	KM analysis: Shorter OS (0.006); Shorter DFS (0.049).
Ramos [18]	MEM-G/2 Patients with invasive ductal BC	Based on ROC-curve analysis	28/45 (62)	Increased LNM	KM analysis: Shorter OS (0.03) Cox multivariate analysis: Shorter OS, HR = 8.8 (0.04)
He [19]	HGY BC patients, random cohort	Absent (0%) and weak (1–25%) staining vs. moderate (25–50%) and strong (>50%) staining	Cohort with available follow-up Weak staining: 42/84 (50) Moderate/ strong staining: 25/84 (30)	Increased tumour size; Increased LNM; Advanced disease stage; Tumour ER over-expression; Tumour PR over-expression.	KM analysis: Shorter OS (0.028) Cox multivariate analysis: Shorter OS, HR = 10.2 (0.006)

The technique used in all mentioned studies for HLA-G detection was immunohistochemistry with formalin-fixed paraffin-embedded tissue. *p*-Values ≤ 0.05 were considered statistically significant. Abbreviations: BC, Breast Carcinoma; CSS, Cancer-Specific Survival; DFS, Disease-Free Survival; ER, Estrogen Receptor; HR, Hazard Ratio; IS, Immune Score; KM, Kaplan–Meier; LNM, Lymph Node Metastasis; ns, not significant; OS, Overall Survival; PR, Progesterone Receptor; RFP, Recurrence-Free Period. * Tissue microarrays were used to determine the percentage of HLA-G expression in the tumour samples.

**Table 2 ijms-22-08265-t002:** Studies on the Correlations between Tumour HLA-G Expression and Clinicopathological Factors and Clinical Outcome of Cervical Carcinoma Patients.

First Author [Ref.]	mAb and Included Patient Cohort	HLA-G Quantification Method	HLA-G+ Samples (%)	Association with Tumour HLA-G Expression	
				Clinico-Pathological Parameters with *p*-Values ≤ 0.05	Clinical Outcome (*p*-Value)
Ferns [22]	4H84 Patients with cervical SCC and AC	Representation of percentage and intensity scores: No expression (0–4) vs. positive expression (5–8)	SSC patients: 23/103 (22) AC patients: 10/33 (31) Total cohort: 33/136 (24)	None declared	KM analysis, in SCC patients with loss of HLA-A expression, *n* = 31: Shorter DFS (0.001); *n* = 30: Shorter CSS (0.004). KM analysis, in patients with loss of classical HLA-class I, *n* = 19: Shorter DFS (0.002); Shorter CSS (0.003). KM analysis, in AC patients: (ns)
* Rodriguez [23]	4H84 CINIII and invasive stage IBI-IVB patients	No expression (=0), focal/weak (=1) or >75% expression (=2) Not defined when a tumour sample was considered as HLA-G positive	16/58 (28)	Decreased HLA-class I expression; IL-10 overexpression.	KM analysis: OS (ns)

The technique used for HLA-G detection was immunohistochemistry with formalin-fixed paraffin-embedded tissue. *p*-Values ≤ 0.05 were considered statistically significant. Abbreviations: AC, Adenocarcinoma; CSS, Cancer-Specific Survival; CIN, Cervical Intraepithelial Neoplasia; DFS, Disease-Free Survival; KM, Kaplan–Meier; ns, not significant; OS, Overall Survival; SCC, Squamous Cell Carcinoma. * Immunohistochemistry analysis was performed on frozen sections.

**Table 3 ijms-22-08265-t003:** Studies on the Correlations between Tumour HLA-G expression and Clinicopathological Factors and Clinical Outcome of Colorectal Carcinoma Patients.

First Author [Ref.]	mAb and Included Patient Cohort	HLA-G Quantification Method	HLA-G+ Samples (%)	Association with Tumour HLA-G Expression	
				Clinico-Pathological Parameters with *p*-Values ≤ 0.05	Clinical Outcome Associated with HLA-G (*p*-Value)
* Reimers [31]	4H84 CRC patients, random cohort	Staining intensity: Weak staining (undetectable to faint staining in <70% of cells) vs. strong staining (weak to moderate staining in >70% of cells)	Weak staining: 350/484 (72) Strong staining: 134/484 (28)	Weak staining: Advanced TNM stage; Increased LNM. Strong staining: Increased number of infiltrating Tregs; Increased HLA-class I expression.	KM analysis: OS (ns); Prolonged DFS (0.040). Cox univariate analysis: OS, HR = 0.76 (ns); Prolonged DFS, HR = 0.75 (0.042). Cox multivariate analysis: OS, HR = 0.88 (ns); DFS, HR = 0.85 (ns).
Cai [25]	4H84 CRC patients, random cohort	Representation of percentage and intensity scores: No expression (0–3) vs. positive expression (4–9)	HLA-G+/ILT4+: 44/88 (50) HLA-G+/ILT4-: 8/88 (9) HLA-G-/ILT4+: 16/88 (18) HLA-G-/ILT4-: 20/88 (23) Total cohort: 52/88 (59)	Advanced TNM stage; Increased ILT4 expression.	KM analysis, HLA-G+/ILT4+ vs. HLA-G-/ILT4-: Shorter OS (0.032) KM analysis, HLA-G+/ILT4+ vs. HLA-G-/ILT4+: Shorter OS (0.043) KM analysis, HLA-G+/ILT4+ vs. HLA-G+/ILT4-: OS (ns)
Zhang [27]	4H84 CRC patients, random cohort	Staining was considered as positive at >5% or >55%	Cohort with available follow-up, >5% staining: 296/417 (71)	More prevalently observed in colon than rectal carcinoma patients	KM analysis: OS (ns) Cox univariate analysis: OS, HR = 1.348 (ns) Cox multivariate analysis: OS, HR = 1.423 (ns)
			Cohort with available follow-up, >55% staining: 273/417 (65)	More prevalently observed in colon than rectal carcinoma patients	KM analysis: Shorter OS (0.042) Cox univariate analysis: Shorter OS, HR = 1.428 (0.044) Cox multivariate analysis: Shorter OS, HR = 1.481 (0.028)
*^,^ ^†^ Kirana [28]	4H84 CRC patients, random cohort.	Staining intensity: No staining vs. moderate or strong staining	Moderate staining: 206/255 (81) Strong staining: 12/255 (5) Total cohort: 218/255 (86)	Strong staining: More prevalently observed in female than male patients	KM analysis, strong vs. no staining, *n* = 48: CSS (ns) KM analysis, strong vs. moderate staining, *n* = 215: Shorter CSS (0.04) KM analysis, strong vs. no/moderate staining, *n* = 251: CCS, HR = 0.571 (ns) KM analysis, moderate vs. no staining, *n* = 239: CSS (ns) KM analysis, strong vs. no/moderate staining in patients with tumour stage II-III, *n* = 167: Shorter CSS (0.01)
^†^ Lin [26]	4H84 CRC patients, random cohort	Any staining >5% was considered as positive	268/379 (71)	Lower TNM stage	KM analysis, *n* = 339: OS (ns) Cox univariate analysis, *n* = 339: OS, HR = 1.267 (ns)
	5A6G7 CRC patients, random cohort	Any staining >5% was considered as positive	229/379 (60)	No association between clinico-pathological variables and HLA-G expression was found	KM analysis, *n* = 339: OS (ns) Cox univariate analysis, *n* = 339: OS, HR = 0.812 (ns)
* Zeestraten [30]	4H84 Only colon carcinoma patients	Any staining is considered as positive (1–100%)	51/251 (20)	No significant correlations	KM analysis: OS (ns); DFS (ns). Cox univariate analysis: OS, HR = 1.2 (ns); DFS, HR = 1.3 (ns).
Guo [29]	MEM-G/2 CRC patients, random cohort	General presence of staining, not further specified	72/102 (71)	Most prevalently observed in adenocarcinoma patients	KM analysis: Shorter OS (0.0243) Cox univariate analysis: Shorter OS, HR = 0.461 (0.029) Cox multivariate analysis: Shorter OS, HR = 0.311 (0.008)
^†^ Ye [24]	HGY CRC patients, random cohort	No staining (0%) vs. weak (1–25%), moderate (25–50%) and strong (>50%) staining	Weak staining: 65/201 (32) Moderate staining: 41/201 (20) Strong staining: 24/201 (12) Total cohort: 130/201 (65)	Advanced TNM stage; Advanced histological grade; Increased tumour depth; Weak immune response; More prevalently observed proximally in colon carcinoma than distally in rectal carcinoma patients.	KM analysis, *n* = 85: Shorter OS (0.001) Cox univariate analysis, *n* = 85: Shorter OS, HR = 6.40 (0.001) Cox multivariate analysis, *n* = 85: Shorter OS, HR = 3.14 (0.021)

The technique used in all mentioned studies for HLA-G detection was immunohistochemistry with formalin-fixed paraffin-embedded tissue. *p*-Values ≤ 0.05 were considered statistically significant. Abbreviations: CRC; Colorectal Carcinoma; CSS, Cancer-Specific Survival; DFS, Disease-Free Survival; HR, Hazard Ratio; KM, Kaplan–Meier; LNM, Lymph Node Metastasis; ns, not significant; OS, Overall Survival; TNM, Tumour, Node, Metastasis. * Tissue microarrays were used to determine the percentage of HLA-G expression in the tumour samples. ^†^ The follow-up data of a limited number of patients was available for the survival analyses.

**Table 4 ijms-22-08265-t004:** Studies on the Correlations between Tumour HLA-G Expression and Clinicopathological Factors and Clinical Outcome of Esophageal Carcinoma Patients.

First Author [Ref.]	mAb and Included Patient Cohort	HLA-G Quantification Method	HLA-G+ Samples (%)	Association with Tumour HLA-G Expression	
				Clinico-Pathological Parameters with *p*-Values ≤ 0.05	Clinical Outcome Associated with HLA-G (*p*-Value)
Lin [32]	4H84 ESCC patients, random cohort	0% (0) vs. 1–25% (1+), 26–50% (2+), 51–75% (3+) or >75% (4+) staining	Cohort with available follow-up: 1+/2+: 14/40 (35) 3+/4+: 9/40 (23) 1+/2+/3+/4+: 23/40 (58)	Advanced TNM stage	KM analysis: Shorter OS (<0.001) Cox univariate analysis: Shorter OS, HR = 3.76 (0.001) Cox multivariate analysis: Shorter OS, HR = 3.83 (0.001) KM analysis, 1+/2+ vs. 0: Shorter OS (0.005) Cox univariate analysis, 1+/2+ vs. 0: Shorter OS, HR = 2.02 (0.01) KM analysis, 3+/4+ vs. 0: Shorter OS (<0.001) Cox univariate analysis, 3+/4+ vs. 0: Shorter OS, HR = 3.02 (<0.001) KM analysis, 3+/4+ vs. 1+/2+: Shorter OS (<0.029)
Zheng [34]	MEM-G/1 ESCC patients, random cohort	<25% vs. >25% staining	42/60 (70)	Advanced differentiation grade; Increased LNM.	KM analysis: Shorter OS (0.01)
Yie [33]	HGY ESCC patients, random cohort	0% (-) vs. 1–25% (1+), 25–50% (2+) or >50% (3+) staining	Cohort with available follow-up: 1+: 27/70 (39) 2+/3+: 32/70 (46) 1+/2+/3+: 59/70 (84)	Advanced tumour grade; Nodal status; Advanced TNM stage; Increased tumour depth; Weak immune response.	KM analysis: Shorter OS (0.001) Cox univariate analysis: Shorter OS, HR = 3.33 (0.001) Cox multivariate analysis: Shorter OS, HR = 2.99 (0.002)

The technique used in all mentioned studies for HLA-G detection was immunohistochemistry with formalin-fixed paraffin-embedded tissue. *p*-Values ≤ 0.05 were considered statistically significant. Abbreviations; CSS, Cancer-Specific Survival; DFS, Disease-Free Survival; ESCC, Esophageal Squamous Cell Carcinoma; HR, Hazard Ratio; KM, Kaplan–Meier; LNM, Lymph Node Metastasis; ns, not significant; OS, Overall Survival; TNM, Tumour, Node, Metastasis.

**Table 5 ijms-22-08265-t005:** Studies on the Correlations between Tumour HLA-G Expression and Clinicopathological Factors and Clinical Outcome of Gastric Carcinoma Patients.

First Author [Ref.]	mAb and Included Patient Cohort	HLA-G Quantification Method	HLA-G+ Samples (%)	Association with Tumour HLA-G Expression	
				Clinico-Pathological Parameters with *p*-Values ≤ 0.05	Clinical Outcome Associated with HLA-G (*p*-Value)
Murdaca [37]	4H84 Gastric adeno-carcinoma patients, random cohort	No staining vs. weak/strong staining	Within stage I patients: 4/14 (29) Within stage II patients: 7/40 (18) Within stage III patients: 13/40 (33) Total cohort: 24/94 (26)	No significant correlations	KM analysis: Shorter OS (<0.0001) Cox proportional hazard analysis: Shorter OS, HR = 4.41 (<0.0001) KM analysis, in stage I patients: OS (ns) KM analysis, in stage II patients: Shorter OS (0.0065) KM analysis, in stage III patients: Shorter OS (<0.0001)
Wan [39]	4H84 GC patients, random cohort	<10% (−) vs. 10–30% (+), 30–50% (++) or >50% (+++) staining	+: 4/49 (8) ++: 17/49 (37) +++: 9/49 (18) Total cohort: 30/49 (61)	Increased preoperative anaemia; Increased tumour depth; Increased LNM; Advanced TNM stage; Decreased number of infiltrating NK cells.	KM analysis: Shorter OS (0.0359); Shorter DFS (0.0438). Cox univariate analysis: Shorter OS, (0.050); DFS (ns). Cox multivariate analysis: OS, 95%CI: 0.500–6.886 (ns); DFS, 95%CI: 0.549–4.307 (ns).
*^, †^ Du [35]	4H84 GC patients, random cohort	0% (−) vs. 1–25% (+), 26–50% (++) or >50% (+++) staining	+: 26/179 (15) ++: 35/179 (20) +++: 28/179 (16) Total cohort: 89/179 (50)	Increased tumour depth; More invaded adjacent organs; Advanced tumour stage; Increased number of infiltrating Tregs.	KM analysis: Shorter OS (<0.001); Shorter DFS (<0.001); *n* = 150 Shorter CSS (<0.001). Cox multivariate analysis: Shorter OS, 95%CI: 1.094–3.040 (0.021); Shorter DFS, 95%CI: 1.187–3.445 (0.010); *n* = 150 Shorter CSS, 95%CI: 1.041–3.192 (0.036).
Ishigami [36]	MEM-G/1 GC patients, random cohort	No staining vs. weak, moderate or strong staining	Weak staining: 16/115 (14) Moderate staining: 19/115 (17) Strong staining: 17/115 (15) Total cohort: 52/115 (45)	Less tumour depth; Decreased LNM; Earlier clinical stage.	KM analysis: Prolonged OS (<0.05)
Yie [40]	HGY GC patients, random cohort	No staining (0%) vs. weak (1–25%), moderate (25–50%) or strong (>50%) staining	Weak staining: 30/160 (19) Moderate staining: 32/160 (20) Strong staining: 51/160 (32) Total cohort: 113/160 (71)	Advanced tumour grade; Increased tumour depth; Increased LNM; Advanced clinical stage; Weak immune response.	KM analysis: Shorter OS (0.001) Cox univariate analysis: Shorter OS, HR = 5.72 (0.0001) Cox multivariate analysis: Shorter OS, HR = 9.08 (0.0001) KM analysis, in patients with disease stage I/II, *n* = 101: Shorter OS (0.001) KM analysis, in patients with disease stage III/IV, *n* = 59: Shorter OS (0.001)
Tuncel [38]	5A6G7 GC patients, random cohort	Any staining >10% was considered as positive	16/52 (31)	Increased LNM; Worse differentiation stage; Tumour type; Advanced TNM stage; Increased number of infiltrating Tregs; Decreased number of CD8+ T cells.	KM analysis: Shorter OS (0.008) Cox univariate analysis: Shorter OS, HR = 3.122 (0.008) Cox multivariate analysis: Shorter OS, HR = 2.662 (0.012)

The technique used in all mentioned studies for HLA-G detection was immunohistochemistry with formalin-fixed paraffin-embedded tissue. *p*-Values ≤ 0.05 were considered statistically significant. Abbreviations; CSS, Cancer-Specific Survival; DFS, Disease-Free Survival; GC, Gastric Carcinoma; HR, Hazard Ratio; KM, Kaplan–Meier; LNM, Lymph Node Metastasis; ns, not significant; OS, Overall Survival; TNM, Tumour, Node, Metastasis. * Tissue microarrays were used to determine the percentage of HLA-G expression in the tumour samples. ^†^ The follow-up data of a limited number of patients was available for the CSS analyses.

**Table 6 ijms-22-08265-t006:** Studies on the Correlations between Tumour HLA-G Expression and Clinicopathological Factors and Clinical Outcome of Hepatocellular Carcinoma Patients.

First Author [Ref.]	mAb and Included Patient Cohort	HLA-G Quantification Method	HLA-G+ Samples (%)	Association with Tumour HLA-G Expression	
				Clinico-Pathological Parameters with *p*-Values ≤ 0.05	Clinical Outcome Associated with HLA-G (*p*-Value)
* Cai [42]	MEM-G/1 HCC patient, random cohort	Mean density calculation as determined by a computerized imaging system	Within early stage HCC patients: 48/76 (63) Total cohort: 99/173 (57)	More prevalently observed in male than female patients; Tregs/CD8+ ratio.	KM analysis: Shorter OS (0.024); RFP (ns). Cox multivariate analysis: Shorter OS, HR = 1.987 (0.004) KM analysis, in patients with early stage HCC: Shorter OS, (0.012); Shorter RFP, (0.038). Cox multivariate analysis, in patients with early stage HCC: Shorter OS, HR 3.145 (0.041); Shorter RFP, HR = 3.208 (0.023).
^†^ Wang [41]	MEM-G/1 HCC patient, random cohort	The appearance of a 39 kDa band corresponding to HLA-G1	24/36 (67)	No significant correlations	KM analysis: Shorter OS (0.027); Shorter RFP (0.035). Cox univariate analysis: Shorter OS, HR = 4.565 (0.044); Shorter RFP, HR = 3.503 (0.048).

*p*-Values ≤ 0.05 were considered statistically significant. Abbreviations; HCC, Hepatocellular Carcinoma; HR, Hazard Ratio; KM, Kaplan–Meier; ns, not significant; OS, Overall Survival; RFP, Recurrence-Free Period. * Tissue microarrays were constructed and immunohistochemistry analysis with formalin-fixed paraffin-embedded tissue was used to determine the percentage of HLA-G expression in the tumour samples. ^†^ Western Blot analysis was used to determine the percentage of HLA-G expression in the tumour samples.

**Table 7 ijms-22-08265-t007:** Studies on the Correlations between Tumour HLA-G Expression and Clinicopathological Factors and Clinical Outcome of Lung Carcinoma Patients.

First Author [Ref.]	mAb and Included Patient Cohort	HLA-G Quantification Method	HLA-G+ Samples (%)	Association with Tumour HLA-G Expression	
				Clinico-Pathological Parameters with *p*-Values ≤ 0.05	Clinical Outcome Associated with HLA-G (*p*-Value)
* Lin [43]	4H84 NSCLC patients, random cohort	0% (0) vs. 1–25% (1), 26–50% (2) or >50% (3) staining	1: 13/101 (13) 2: 16/101 (16) 3: 13/101 (13) Total cohort: 42/101 (42)	Advanced disease stage	KM analysis, *n* = 51: OS (ns)
Zhang [46]	4H84 NSCLC patients, random cohort	Representation of percentage and intensity scores: No expression (<4) vs. positive expression (≥4)	HLA-G+/ILT4+: 29/81 (36) HLA-G+/ILT4-: 13/81 (16) HLA-G-/ILT4+: 9/81 (11) HLA-G-/ILT4-: 30/81 (37) Total cohort: 42/81 (52)	Increased LNM; Advanced disease stage; Worse differentiation stage; ILT4 overexpression.	KM analysis, HLA-G+/ILT4+ vs. HLA-G-/ILT4+: Shorter OS (0.021) KM analysis, HLA-G+/ILT4+ vs. HLA-G-/ILT4-: Shorter OS (0.048) KM analysis, HLA-G+/ILT4+ vs. HLA-G+/ILT4-: OS (ns)
Yie [45]	HGY NSCLC patients, random cohort	0% (−) vs. 1–25% (+), 26–50% (++) and >50% (+++) staining	Percentage of HLA-G-positive samples within specific staining groups was unattainable Total cohort with available follow-up: 23/39 (59)	Increased LNM; Advanced disease stage; Weak immune response.	KM analysis: Shorter OS (0.001) Cox univariate analysis: Shorter OS, HR = 4.01 (0.003) Cox multivariate analysis: Shorter OS, HR = 4.09 (0.010)
* Yan [44]	5A6G7 NSCLC patients, random cohort	Any staining >5% was considered as positive	SSC patients: 4/66 (6) AC patients: 40/55 (73) ASC patients: 1/10 (10) Total cohort: 41/123 (34)	More prevalently observed in adenocarcinoma than squamous or adenosquamous carcinoma patients; More prevalently observed in female than male patients.	Cox univariate analysis: OS, HR = 1.15 (ns) KM analysis, in SSC patients, *n* = 62: OS (ns) Cox univariate analysis, in SSC patients, *n* = 62: OS, HR = 2.76 (ns) Cox multivariate analysis, in SSC patients, *n* = 62: OS, HR = 4.05 (ns) KM analysis, in AC patients, *n* = 51: OS (ns) Cox univariate analysis, in AC patients, *n* = 51: OS, HR = 1.04 (ns) Cox univariate analysis, in ASC patients: OS, HR = 0.03 (ns)

The technique used in all mentioned studies for HLA-G detection was immunohistochemistry with formalin-fixed paraffin-embedded tissue. *p*-Values ≤ 0.05 were considered statistically significant. Abbreviations; AC, Adenocarcinoma; ASC, Adenosquamous Carcinoma; HR, Hazard Ratio; KM, Kaplan–Meier; LNM, Lymph Node Metastasis; ns, not significant; NSCLC, Non-Small-Cell Lung Carcinoma; OS, Overall Survival; SSC, Squamous Cell Carcinoma. * The follow-up data of a limited number of patients was available for the survival analyses.

**Table 8 ijms-22-08265-t008:** Studies on the Correlations between Tumour HLA-G Expression and Clinicopathological Factors and Clinical Outcome of Oral Carcinoma Patients.

First Author [Ref.]	mAb and Included Patient Cohort	HLA-G Quantification Method	HLA-G+ Samples (%)	Association with Tumour HLA-G Expression	
				Clinico-Pathological Parameters with *p*-Values ≤ 0.05	Clinical Outcome Associated with HLA-G (*p*-Value)
Imani [48]	4H84 Oral SCC patients, random cohort	IRS (representation of percentage and intensity): No expression (0) vs. low (≤2) vs. high (≥2) expression	0: 0/33 (0%) ≤2: 6/33 (18%) ≥2: 27/33 (82%)	Advanced tumour stage; Increased LNM; Increased distant metastasis.	KM analysis, groups compared unverifiable: OS (ns) Spearman’s CC: OS, CC = −0.374 (0.018)
Goncalves [47]	MEM-G/2 Oral SCC patients, random cohort	IRS (representation of percentage and intensity): No expression (0) vs. low (≤2) or high (≥2) expression	0: 0/60 (0) ≤2: 30/60 (50) ≥2: 30/60 (50)	Increased tumour depth	KM analysis: OS (ns)
Mosconi [49]	MEM-G/2 Intraoral muco-epidermoid carcinoma patients, random cohort	Low expression (<50% staining) vs. high expression (>50% staining)	Unverifiable	Advanced histological grade	KM analysis, *n* = 30: OS (ns)

The technique used in all mentioned studies for HLA-G detection was immunohistochemistry with formalin-fixed paraffin-embedded tissue. *p*-Values ≤ 0.05 were considered statistically significant. Abbreviations; CC, Correlation Coefficient; HR, Hazard Ratio; IRS; Immune Reactivity Scores; KM, Kaplan–Meier; n/a, not applicable; ns, not significant; OS, Overall Survival; SCC Squamous Cell Carcinoma.

**Table 9 ijms-22-08265-t009:** Studies on the Correlations between Tumour HLA-G Expression and Clinicopathological Factors and Clinical Outcome of Ovarian Carcinoma Patients.

First Author [Ref.]	mAb and Included Patient Cohort	HLA-G Quantification Method	HLA-G+ Samples (%)	Association with Tumour HLA-G Expression	
				Clinico-Pathological Parameters with *p*-Values ≤ 0.05	Clinical Outcome Associated with HLA-G (*p*-Value)
* Rutten [53]	4H84 Type II, high grade ovarian carcinoma patients	Representation of percentage and intensity; Normal expression (<3) vs. upregulated expression (>3)	81/152 (53)	More residual tumour after debulking surgery; Increased platinum sensitivity.	KM analysis: Prolonged OS (0.001); Prolonged CSS (0.008); Prolonged PFS (0.036). Cox univariate analysis: Prolonged CSS, HR = 1.69 (0.009) Cox multivariate analysis: Prolonged CSS, HR = 1.62 (0.020) KM analysis, in patients with tissue collected prior to chemotherapy, *n* = 108: Prolonged CSS (0.011); Prolonged PFS (0.027). KM analysis, in patients with downregulated HLA-A, *n* = 137: CSS (ns)
^†^ Babay [51]	4H84 Ovarian carcinoma patients, random cohort	<1% staining (0) vs. 1–5% (1), 6–25% (2), 26–50% (3) or >50% (4) staining	Percentage of HLA-G-positive samples within specific staining groups was unattainable Cohort with available follow-up: 36/51 (71)	No significant correlations	KM analysis: OS (ns) Multivariate binomial logistic regression analysis: Recurrence, HR = 4.115 (ns)
Jung [52]	4H84 Ovarian carcinoma patients, random cohort	Mild (0–25%; 1+), moderate (25–50%; 2+) and strong (>50%; 3+) staining Optimal cut-off value was determined by ROC curve analysis at 17% for HLA-G detection	1+: 22/40 (55) 2+: 8/40 (20) 3+: 10/40 (25) > 17%: 24/40 (60)	Advanced disease stage	KM analysis: Shorter OS (0.04); PFS (ns). Cox univariate analysis: Shorter OS, HR = 3.00 (0.04) Cox multivariate analysis: OS (ns) KM analysis, in patients with specific cancer stages: OS (ns)
	^‡^ 4H84 Ovarian carcinoma patients, random cohort	Optimal cut-off value was determined by ROC curve analysis at DV 1.14 for HLA-G detection	DV >1.14: 18/40 (45)	Advanced disease stage; CA125 over-expression.	KM analysis: OS (ns); PFS (ns). Cox univariate analysis: OS, HR = 1.48 (ns)
Andersson [50]	MEM-G/1 Advanced stage III/IV, serous ovarian adeno-carcinoma patients	No staining (0) vs. 1–25% (1), 26–50% (2), 51–75% (3) or >75% (4) staining	Percentage of HLA-G-positive samples within specific staining groups was unattainable Total cohort: 14/72 (20)	Absence of infiltrating Tregs and CD8+ T cells	KM analysis: OS (ns) KM analysis, in patients with HLA-A*02, without CD8+ cells and with HLA-G expression vs. patients with HLA-A otherwise, presence of CD8+ cells and without HLA-G expression, *n* = 42: Shorter OS (0.006)
Zhang [54]	5A6G7 Ovarian carcinoma patients, random cohort	Any staining >5% was considered as positive	Cohort with available follow-up: 14/17 (82)	No significant correlations	KM analysis: OS (ns) Cox univariate analysis: OS, HR = 0.58 (ns) Cox multivariate analysis: OS, HR = 0.48 (ns)

The technique used for HLA-G detection was immunohistochemistry with formalin-fixed paraffin-embedded tissue unless indicated otherwise. *p*-Values ≤ 0.05 were considered statistically significant. Abbreviations; CSS, Cancer-Specific Survival; DFS, Disease-Free Survival; DV, Densitometer Value; HR, Hazard Ratio; KM, Kaplan–Meier; n/a, not applicable; ns, not significant; OS, Overall Survival; PFS, Progression-Free Survival; RFP, Recurrence-Free Period. * Tissue microarrays were used to determine the percentage of HLA-G expression in the tumour samples. ^†^ The follow-up data of a limited amount of patients was available for the survival analyses. ^‡^ Western Blot analysis was used to determine the percentage of HLA-G expression in the tumour samples.

**Table 10 ijms-22-08265-t010:** Studies on the Correlations between Tumour HLA-G Expression and Clinicopathological Factors and Clinical Outcome of Pancreatic Carcinoma Patients.

First Author [Ref.]	mAb and Included Patient Cohort	HLA-G Quantification Method	HLA-G+ Samples (%)	Association with Tumour HLA-G Expression	
				Clinico-Pathological Parameters with *p*-Values ≤ 0.05	Clinical Outcome Associated with HLA-G (*p*-Value)
Hiraoka [56]	4H84 PDAC patients, random cohort	Any staining >5% was considered as positive	36/98 (37)	No significant correlations	KM analysis: Shorter OS (0.005); Shorter DFS (0.009). Cox univariate analysis: Shorter OS, HR = 2.026 (0.006); Shorter DFS, HR = 1.867 (0.011). Cox multivariate analysis: Shorter OS, HR = 1.824 (0.021); Shorter DFS, HR = 1.828 (0.015).
* Sideras [57]	MEM-G/1 Pancreatic and ampullary carcinoma patients	Any staining was considered as positive Lowest −2 log likelihood was chosen as cut-off value for survival analyses	Percentages of HLA-G-positive samples within pancreatic (*n* = 148) and ampullary (*n* = 76) carcinoma patient groups was unattainable Cohort with available follow-up: 32/217 (15)	More peri-neural invasion; Increased HVEM expression.	KM analysis: Prolonged OS (0.004); Prolonged DFS (0.008). Cox univariate analysis: Prolonged CSS, HR = 0.43 (0.004); Prolonged DFS, HR = 0.51 (0.008). Cox multivariate analysis: CSS, HR = 0.53 (ns) Cox univariate analysis, in patients with pancreas carcinomas: CSS, HR = 0.66 (ns) Cox univariate analysis, in patients with ampulla carcinomas: Prolonged CSS, HR = 0.38 (0.021)
* Zhou [59]	None specified Pancreatic carcinoma patients, random cohort	Negative (<5%) or local (5–75%) staining vs. diffuse (>75%) staining	Percentage of HLA-G-positive samples with available follow-up data with local staining was unattainable Cohort with available follow-up, with diffuse staining: 20/143 (14)	Advanced tumour stage; Decreased TIL number.	KM analysis: Shorter OS (<0.001) Cox univariate analysis: Shorter OS (<0.001) Cox multivariate analysis: Shorter OS, HR = 2.135 (0.011)
^†^ Xu [58]	Polyclonal Rabbit Ab PDAC patients, cohort uncertain	Sum of proportion and intensity	78/122 (64)	Advanced TNM; Increased LNM; Worse differentiation.	Cox multivariate analysis, *n* = unknown: Shorter OS, HR = 3.894 (<0.001)

The technique used in all mentioned studies for HLA-G detection was immunohistochemistry with formalin-fixed paraffin-embedded tissue. *p*-Values ≤ 0.05 were considered statistically significant. Abbreviations; CSS, Cancer-Specific Survival; DFS, Disease-Free Survival; HR, Hazard Ratio; HVEM, Herpesvirus Entry Mediator; KM, Kaplan–Meier; LNM, Lymph Node Metastasis; n/a, not applicable; ns, not significant; OS, Overall Survival; PDAC, Pancreatic Ductal Adenocarcinoma; TIL, Tumour Infiltrating Lymphocyte; TNM, Tumour, Node, Metastasis. * Tissue microarrays were used to determine the percentage of HLA-G expression in the tumour samples. ^†^ No full text was available.

**Table 11 ijms-22-08265-t011:** Studies on the Correlations between Tumour HLA-G Expression and Clinicopathological Factors and Clinical Outcome of Residual Carcinoma Type Patients.

First Author [Ref.]	mAb and Included Patient Cohort	HLA-G Quantification Method	HLA-G+ Samples (%)	Association with Tumour HLA-G Expression	
				Clinico-Pathological Parameters with *p*-Values ≤ 0.05	Clinical Outcome Associated with HLA-G (*p*-Value)
*^,†^ Bijen [60]	4H84 Endo-metrial carcinoma patients, random cohort	Representation of percentage and intensity: No expression (≤2.5) vs. low (2.5–6.5) or strong (≥6.5) expression	Percentage of HLA-G-positive samples within specific scoring groups was unattainable Total cohort: 209/525 (40)	HLA-class I overexpression	Cox univariate analysis, *n* = 111: DFS, HR = 0.85 (ns); *n* = 73 CSS, HR = 1.01 (ns). Cox univariate analysis, in patients with cancer stage type I, *n* = 71: DFS, HR = 0.80 (ns); *n* = 35 CSS, HR = 0.81 (ns). Cox univariate analysis, in patients with cancer stage type II, *n* = 40: DFS, HR = 0.86 (ns); *n* = 38 CSS, HR = 1.08 (ns).
Lopes [61]	MEM-G/2 LSCC patients, random cohort	0% (0) and 1–25% (1) staining vs. 26–50% (2) and >50% (3) staining	Percentage of HLA-G-positive samples within specific staining groups was unattainable All LSCC samples showed HLA-G staining: 40/40 (100)	Distant metastasis	KM analysis: OS (ns)
De Figueiredo-Feitosa [62]	5A6G7 PTC and FTC patients, random cohort	No (0%, -), mild (1–25%, +) or moderate (26%-50%, ++) expression vs. strong (>50%, +++) expression	PTC patients: +: 2/72 (3) ++: 12/72 (17) +++: 58/72 (80)	Increased tumour size	KM analysis: DFS (ns)
			FTC patients: +: 1/19 (5) ++: 3/19 (16) +++: 15/19 (79)	No significant correlations	n/a
* Jasinski-Bergner [63]	4H84 RCC patients, random cohort	IRS (product of percentage and intensity). Staining intensity; negative (0), weak (1), moderate (2) and strong (3) Not defined when a tumour sample was considered HLA-G positive	Percentage of HLA-G-positive samples within specific IRS groups was unattainable Total cohort: 186/367 (51)	Strong cytoplasmic HLA-G expression: Advanced tumour grade	KM analysis: OS (ns)

The technique used for HLA-G detection in all mentioned studies was immunohistochemistry with formalin-fixed paraffin-embedded tissue. *p*-Values ≤ 0.05 were considered statistically significant. Abbreviations; CCS, Cancer-Specific Survival; DFS, Disease-Free Survival; FTC, Follicular Thyroid Carcinoma; HR, Hazard Ratio; IRS, Immunoreactive Score; KM, Kaplan–Meier; LSCC, Lip Squamous Cell Carcinoma; n/a, not applicable; ns, not significant; OS, Overall Survival; PTC, Papillary Thyroid Carcinoma. * Tissue microarrays were used to determine the percentage of HLA-G expression in the tumour samples. ^†^ The follow-up data of a limited number of patients was available for the survival analyses.

## Data Availability

Not applicable.

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
