# Peer review of "A Critical Assessment of the Association between HLA-G Expression by Carcinomas and Clinical Outcome"

_ijms, 2021, doi:10.3390/ijms22158265_

Round 1
Reviewer 1 Report
Water et al. provides a critical assessment of works of literature on the association between HLA-G expression and clinical outcome of patients with different kinds of cancer. The authors correctly pointed out the methodological variations among studies consistent with the heterogeneous outcome of the results. In the end, the authors established that role of HLA-G as a novel immune checkpoint molecule in cancer is premature. Thus, future research is necessary to understand the role of HLA-G in cancer. The review provides sufficient references and background. The tables can be improved to enhance readability.
Minor comments:
- Line 734: “It may by possible …” -> “It may be possible …”
Author Response
- Water et al. provides a critical assessment of works of literature on the association between HLA-G expression and clinical outcome of patients with different kinds of cancer. The authors correctly pointed out the methodological variations among studies consistent with the heterogeneous outcome of the results. In the end, the authors established that role of HLA-G as a novel immune checkpoint molecule in cancer is premature. Thus, future research is necessary to understand the role of HLA-G in cancer. The review provides sufficient references and background. The tables can be improved to enhance readability.
First and foremost, thank you for taking the time to read our manuscript. It is greatly appreciated.
We agree that there is a surplus of information per box to take in from the tables and that this might go at the expense of the readability of the tables in general. However, the goal (and novelty) of this review was to provide an objective and, above all, complete overview on all studies discussing the association between tumour HLA-G expression and clinical patient outcome. Although we appreciate your suggestion, the makeup and contents of the tables have been discussed at length within our own research group and after many iterations we believe that the optimal balance between completeness, conciseness and clearness has been reached in the current form.
Furthermore, during translation of the manuscript to the IJMS layout tables have shifted slightly and show a less optimal layout. In an effort to improve the overall format, we turned off automatic hyphenation (we did, however, not manage to do this for the tables exclusively) and reduced the letter size from 10 to 9 to make the text fit better in the boxes, while maintain IJMS table measure standards. Maybe the team of IJMS can make further improvement to the overall layout of the tables, while maintaining the content.
- Line 734: “It may by possible …” -> “It may be possible …”
We thank the reviewer for pointing this out. We have revised the error.
Reviewer 2 Report
1) L 24-28. Abstract. Overall, HLA-G expression did not univocally result in poor clinical outcome of carcinoma patients. This implies that tumour HLA-G expression is not necessarily part of an inhibited tumour-immune response and tumour progression. Consequently, it remains elusive whether HLA-G expression by carcinomas functions as an immune checkpoint molecule affecting a tumour-immune response. It may also reflect derailed control of gene expression in tumours, with no real functional consequences. Please underline the aim of the article.
2) L 60-74. Introduction. The aim of this review was to uncover to what extend it is justified that HLA-G ex- 60 pression is considered as a target for ICI therapy in carcinomas. To this end, a comprehensive and objective overview concerning the association between HLA-G expression in carcinomas and clinical outcome of patients is provided. This includes the methods and types of analyses that were used to determine tumour HLA-G expression and its clinical impact. Several measures were taken to be able to better compare the various published studies among each other. Included studies were restricted to carcinomas and categorised by anatomical site to gain uniformity. Both, statistically significant as well as non-significant analyses concerning the association between tumour HLA-G expression and clinical patient outcome were included to obtain the most complete picture on this association as possible. Additionally, the used HLA-G-detecting mAbs and HLA-G quantification methods were considered when interpreting the results from the different studies. Lastly, not only clinical outcome, but all statistically significant clinicopathological patient and tumour characteristics associated with HLA-G expression were included. In this way, potential patterns between HLA-G expression and tumour development in carcinoma patients were uncovered. Could you please underline the novelty of your manuscript?
3) Introduction. Please underline the methods and types of analyses used to evaluate the data.
4) L 76 2. Association Between HLA-G Expression and Clinicopathological Parameters in Carcinoma Patients 2.1. Breast Carcinoma. In summary, HLA-G quantification methods varied between the studies, which partly accounts for the variability in the range in percentage of reported HLA-G-positive tumour samples. Nonetheless, all studies demonstrated significantly poor clinical outcome of patients associated with higher levels of tumour HLA-G expression, either as a single parameter or in conjunction with other immune parameters [17-21]. Also, clinicopathological tumour characteristics indicative for poor disease prognosis were associated with HLA-G expression. This shows, HLA-G associates in many respects with poor clinical outcome in breast carcinomas. Could you please ameliorate this paragraph?
5) L 238 2.3. Colorectal Carcinoma. In conclusion, although several studies demonstrated that tumour HLA-G expression was associated with poor clinal outcome, other studies that yield more statistical power due to larger sample size did not observe significant associations between tumour HLA-G expression and clinical outcome of CRC patients [24-26, 29, 30]. These differences in the association between tumour HLA-G expression and clinical outcome of patients between the studies cannot be explained based on variations in HLA-G quantification methods or reported percentages of HLA-G-positive tumours. Two other studies showed mixed results concerning the HLA-G-related clinical outcome of patients [27, 28]. Reimers, et al. even reported prolonged DFS time in patients with upregulated HLA-G expression and, therefore, concluded that HLA-G may have protective effects in CRC patients [31]. Thus, based on the currently provided data on the association between tumour HLA-G expression and clinical outcome, the role of HLA-G in CRC patients is not clear and requires further attention. Please underline the most important information.
6) 3. Discussion. Another factor that should be highlighted in combination with HLA-G expression in tumours is the presence of infiltrating immune cells. Please add this recent reference:
- a) Ruaro B, Salton F, Braga L, Wade B, Confalonieri P, Volpe MC, Baratella E, Maiocchi S, Confalonieri M. The History and Mystery of Alveolar Epithelial Type II Cells: Focus on Their Physiologic and Pathologic Role in Lung. Int J Mol Sci. 2021 Mar 4;22(5):2566. doi: 10.3390/ijms22052566.
7) 4. Conclusion In summary, in this review a critical assessment was made on the association between HLA-G expression and clinical outcome of patients with carcinomas. Overall, discrepancies in methodological approaches of HLA-G-related studies hampered a proper comparison of these studies with each other. Also, considerable heterogeneity in the association between HLA-G expression and clinical patient outcome was observed between the various studies, which may partly be caused by variations in methodologies. Nonetheless, it appeared that the role HLA-G expression plays, differs according to the anatomical origin of the carcinoma. HLA-G expression in breast, gastric, hepatocellular and esophageal carcinoma patients was associated with significantly poor clinical outcome. HLA-G might therefore have a functional role in these carcinoma types and thus might act as an immune checkpoint molecule. Cervical, colorectal, lung, oral, ovarian and pancreatic carcinoma patients did not univocally reveal poor clinical outcome associated with HLA-G expression. It therefore remains uncertain what the exact role of HLA-G expression is in these carcinoma types. Furthermore, before any therapeutic application can be applied that targets HLA-G, more information is needed on tumour expression of HLA- G and of its isoforms. The development of novel and more specific HLA-G-detecting mAbs is emphasised. Also, future research should focus on the role of HLA-G in tumourimmune interactions, as this is a multifactorial and complex process. In our opinion, the declaration of HLA-G as novel immune checkpoint molecule in cancer is premature. Please summarise the most important information.
Reviewer 3 Report
This manuscript deals with a deep analysis of studies on HLAG expression in carcinomas. The scheme of this work is good to follow for each kind of carcinoma the relevance of the expression of HLAG and its possible association with clinical outcome. After reading this review, I would say that the main message on HLAG is that its relevance in carcinoma is quite questionable. Indeed, starting from the use of antibodies that do not recognize only HLAG or that can react differently with different isoforms of HLAG, it is clear that we are far from a definitive answer to whether HLAG is relevant or not for carcinoma development and growth.
On the other hand, no information on this topic using some other different methodological approach avoiding the use of anti-HLAG antibodies (with a not so well-defined reactivity and specificity) is considered and analysed. For instance, is it possible to analyse bioinformatic data for the mRNA expression of HLAG and HLAG isoforms? This hypothesis can be suggested if it is not applicable and why.
Furthermore, the relevance of soluble molecules of HLAG in carcinomas that some studies consider of a certain relevance on this topic are not considered.
I understand well that the scope of this review was to analyse the possible correlation between HLAG expression and carcinoma, but from this review I remain with the idea that the correct analysis of HLAG expression is too elusive to give a definitive answer on the topic.
The hypothesis to use HLAG as a therapeutic target seems too far because one of the main tool to target specific molecule such as humanized monoclonal antibodies apparently do not exist.
In conclusion, I would consider this work more appropriate for its content to a scientific journal on cancer instead of IJMS because no clear insight on the molecular features of HLAG are reported.
The presence of several tables can aid the reader, but I would say that these tables should be more synthetic instead of repeating some information already listed in the text. Perhaps, also, a conclusion of the significance of the given study should be indicated to summarize the relevance of that study.
Perhaps, some figures depicting the data reported can aid the reader to take home a clear message on the topic.
Author Response
This manuscript deals with a deep analysis of studies on HLAG expression in carcinomas. The scheme of this work is good to follow for each kind of carcinoma the relevance of the expression of HLAG and its possible association with clinical outcome. After reading this review, I would say that the main message on HLAG is that its relevance in carcinoma is quite questionable. Indeed, starting from the use of antibodies that do not recognize only HLAG or that can react differently with different isoforms of HLAG, it is clear that we are far from a definitive answer to whether HLAG is relevant or not for carcinoma development and growth.
Thank you for your kind words regarding our manuscript. We can tell by your comment that our main message has been successfully conveyed and you read the manuscript attentively. For that, our sincerest gratitude.
On the other hand, no information on this topic using some other different methodological approach avoiding the use of anti-HLAG antibodies (with a not so well-defined reactivity and specificity) is considered and analysed. For instance, is it possible to analyse bioinformatic data for the mRNA expression of HLAG and HLAG isoforms? This hypothesis can be suggested if it is not applicable and why.
Thank you for your suggestion. Indeed, the current use of anti-HLA-G mAbs is suboptimal, as also described in our manuscript. However, for the scope of this review, we have limited ourselves to the direct expression of HLA-G proteins in tumour tissues. We feel that more upstream indications of presence of HLA-G (e.g. through mRNA expression) is not a fair reflection of the actual expression of HLA-G on protein level. It has been shown that HLA-G mRNA levels, HLA-G gene DNA methylation status and HLA-G expression levels did not correspond in colorectal tumour tissues [Swets, et al. 2016]. This is because HLA-G mRNA is under strict post-translational control by, amongst other things, miRNA’s [Krijgsman, et al. 2020]. Nonetheless, we do agree that it is worth mentioning. Therefore we have added the following paragraph in the discussion section:
“Other approaches to measure HLA-G in tumour lesions could include using HLA-G mRNA expression levels. However, it has been shown in CRC samples that HLA-G mRNA expression levels does not necessarily translate to HLA-G protein expression [71, 72]. This is because HLA-G mRNA is under strict post-translational control by HLA-G mRNA-specific miRNA’s [63, 73-75]. Therefore, more upstream indications of presence of HLA-G in tumour lesions (e.g. through mRNA expression) is not a fair reflection of the actual expression of HLA-G on protein level.” L 643 – 649.
Furthermore, the relevance of soluble molecules of HLAG in carcinomas that some studies consider of a certain relevance on this topic are not considered.
We agree that soluble HLA-G isoforms may play an important role in the general inhibition of the tumour-immune response and should therefore be investigated more intensively. However, it did not fall under the scope of the current review. In this review, we focused on the direct expression of HLA-G molecules by the tumour. Studies investigating the association between soluble HLA-G and clinical outcome of carcinoma patients often used patients’ serum. Therefore, the majority of these studies did not fall under the scope of the current review. Nonetheless, if studies investigated soluble HLA-G expression in tumour lesions (for instance by using 5A6G7 mAbs that only recognise soluble HLA-G molecules), we did include these studies. Furthermore, circulating soluble HLA-G expression levels does not reflect local, tumour HLA-G expression within individual lung carcinoma patients, as described by Lin, et al. (2010).
I understand well that the scope of this review was to analyse the possible correlation between HLAG expression and carcinoma, but from this review I remain with the idea that the correct analysis of HLAG expression is too elusive to give a definitive answer on the topic.
The hypothesis to use HLAG as a therapeutic target seems too far because one of the main tool to target specific molecule such as humanized monoclonal antibodies apparently do not exist.
In conclusion, I would consider this work more appropriate for its content to a scientific journal on cancer instead of IJMS because no clear insight on the molecular features of HLAG are reported.
Thank you for your suggestion. We have decided, in consultation with the IJMS editorial board, to continue to get the manuscript processed for IJMS, especially since the designated special issue concerns HLA-G.
The presence of several tables can aid the reader, but I would say that these tables should be more synthetic instead of repeating some information already listed in the text. Perhaps, also, a conclusion of the significance of the given study should be indicated to summarize the relevance of that study.
We acknowledge that the tables in the current form do not recapitulate the relevance of corresponding studies. We have reconciliated your proposal to also provide this additional information. However, one of the major goals of this review was to remain as objective as possible. We believe that by assigning relevance to studies a certain degree of subjectivity comes into play through our value of judgement. Therefore, we have decided not to add this information to the tables and discuss in the body text only.
Perhaps, some figures depicting the data reported can aid the reader to take home a clear message on the topic.
We agree, and, preferably, we would have liked to be able to summarize all available data in a comprehensive figure. However, all compared studies within the different carcinoma types diverge significantly in both methods, as well as results. Therefore, to merge data derived from different studies into one figure would not be correct. Besides, we must emphasise that the current review is a description of the currently available literature and not a meta-analysis. Thus, although we consider your opinion to be of the utmost importance, we have decided not take any further action.
Round 2
Reviewer 2 Report
The manuscript has been improved as required. I have no further comments.
Reviewer 3 Report
The Authors have replied to the large majority of concerns raised with a written reply without making substantial changes of the manuscript as requested. Although, I understand well the authors' point of view, I remain of my specific opinion on this work that it is well performed and assembled, but the reader will get from it that HLA-G is something of quite elusive in cancer and a comprehensive methodological approach using not only immunohistochemistry with not well-defined specificity against HLA-G should be applied.
I do not have any against the endorsement for publication of this work but actually, I cannot accept it as it stands.